# Molecular Mechanisms of Action of Eugenol in Cancer: Recent Trends and Advancement

**DOI:** 10.3390/life12111795

**Published:** 2022-11-06

**Authors:** Ipsa Padhy, Paramita Paul, Tripti Sharma, Sabyasachi Banerjee, Arijit Mondal

**Affiliations:** 1Department of Pharmaceutical Chemistry, School of Pharmaceutical Sciences, Siksha ‘O’Anusandhan (Deemed to be University), Bhubaneswar 751003, Odisha, India; 2Department of Pharmaceutical Technology, University of North Bengal, Raja Rammohunpur 734013, West Bengal, India; 3Department of Pharmaceutical Chemistry, Gupta College of Technological Sciences, Asansol 713301, West Bengal, India; 4Department of Pharmaceutical Chemistry, M. R. College of Pharmaceutical Sciences and Research, Balisha 743234, West Bengal, India

**Keywords:** eugenol, dietary phytochemicals, anticancer, signaling pathways, drug delivery, therapeutic profile

## Abstract

Background: Cancer is, at present, among the leading causes of morbidity globally. Despite advances in treatment regimens for cancer, patients suffer from poor prognoses. In this context, the availability of vast natural resources seems to alleviate the shortcomings of cancer chemotherapy. The last decade has seen a breakthrough in the investigations related to the anticancer potential of dietary phytoconstituents. Interestingly, a handsome number of bioactive principles, ranging from phenolic acids, phenylpropanoids, flavonoids, stilbenes, and terpenoids to organosulphur compounds have been screened for their anticancer properties. Among the phenylpropanoids currently under clinical studies for anticancer activity, eugenol is a promising candidate. Eugenol is effective against cancers like breast, cervical, lung, prostate, melanomas, leukemias, osteosarcomas, gliomas, etc., as evident from preclinical investigations. Objective: The review aims to focus on cellular and molecular mechanisms of eugenol for cancer prevention and therapy. Methods: Based on predetermined criteria, various scholarly repositories, including PubMed, Scopus, and Science Direct were analyzed for anticancer activities of eugenol. Results: Different biochemical investigations reveal eugenol inducing cytotoxicity, inhibiting phases of the cell cycles, programmed cell death, and auto-phagocytosis in studied cancer lines; thus, portraying eugenol as a promising anticancer molecule. A survey of current literature has unveiled the molecular mechanisms intervened by eugenol in exercising its anticancer role. Conclusion: Based on the critical analysis of the literature, eugenol exhibits vivid signaling pathways to combat cancers of different origins. The reports also depict the advancement of novel nano-drug delivery approaches upgrading the therapeutic profile of eugenol. Therefore, eugenol nanoformulations may have enormous potential for both the treatment and prevention of cancer.

## 1. Introduction

Cancer has been one of the daunting causes of high morbidity in the human population for the past two decades. Systematic cancer research has revealed that the growth of malignant cancers with the ability to metastasize is caused by a complex collection of events, including the rapid spread and unregulated proliferation of aberrant cells [1]. Radiations, chemical substances, viruses, bacteria, genetic abnormalities, mutations, defective genes, lack of tumor-suppressive genes, defective cell cycle, apoptotic machinery, etc., have been found to majorly trigger carcinogenesis [2]. Although there are many different cancer therapy options available, some of them may be unsuccessful due to negative side effects or increased resistance to traditional anticancer medicines [3].

Moreover, conventional cancer therapies are found to affect healthy cells as well which leads to several unnecessary risks. Hence, natural compounds, that are having potential anticancer activity in vitro, could serve as a better alternative to the researchers as it regulates a wide range of cellular activities, namely cellular growth and differentiation, metastasis, apoptosis, DNA damage, and repair [4]. Natural compounds are a valuable asset in the research and development of new medications, primarily for cancer therapy as they have the benefits of having lesser toxicity and are easily tolerated by the human body at a higher dose compared to conventional chemotherapeutic drugs [5]. More than 5000 phytochemicals are reported in literature having antineoplastic properties which might offer vital resources for developing novel anti-cancer agents, and a safer alternative to a variety of synthetic medicines presently utilized in clinical treatments [6].

Amongst the natural compounds, essential oils from dietary plants and aromatic herbs play an important role in cancer therapy [7]. Eugenol, chemically allyl side chain guaiacol, is a natural monoterpene that belongs to the class of phenylpropanoids. It is a principal component of essential oils that are derived from basil, cinnamon, bay leaf, nutmeg, and clove, among which clove flower buds are the principal source. It acts as an anti-inflammatory [8], analgesic [9], antioxidant [10], antifungal [11], antimicrobial [12], antiviral [13], and antileishmanial [14] among other pharmacological properties. It has profound applications in the food, beverage, cosmetic, and pharma industry for its aroma [15,16]. Of note, eugenol has been evaluated as a versatile therapeutic molecule underscoring its anticancer activity [17]. Eugenol and its synthesized derivatives were reported to have an anti-proliferative effect against cancers of various anatomical origins. In vivo and in vitro investigations envisage specific molecular mechanistic pathways of antiproliferative activity of eugenol, like induction of apoptosis and cell cycle arrest [18,19].

Just a few previous reviews offer detailed knowledge of the progress of this significant research area. Previous review articles present a brief overview of the compilation and exploration of the therapeutic properties of eugenol in cancer [20,21]. There are no publications on the overview of novel formulations of eugenol in cancer. One review focuses on the relationship of eugenol in cancer therapy with prooxidants and antioxidant activities [22]. Although there is copious information on the anticancer properties of eugenol based on in vivo and in vitro studies, a comprehensive assessment of the anticancer properties of eugenol is yet to be performed. Keeping this in mind, our current work elucidates the anticancer potential of eugenol in different types of cancer, highlights underlying mechanisms, and emphasizes various nanoformulations enhancing its anticancer property as well as upgrading its bioavailability. 

## 2. Methodology for Literature Search and Selection of Anticancer Studies

The Preferred Reporting Items for Systematic Reviews and Meta-Analysis (PRISMA) criteria [23], the recommended standard for systematic reviews, was followed for this work. Electronic databases and search engines such as Scopus, PubMed, Google Scholar, and Science Direct were used to gather relevant literature furnishing data regarding the anticancer activity of eugenol. There were no time restraints for this search and the last search was performed in September 2022. Various keywords, such as eugenol, hematological cancer, breast cancer, liver cancer, colon cancer, stomach cancer, prostate cancer, skin cancer, lung cancer, leukemia, anticancer, antitumor, prevention, apoptosis, in vitro, and in vivo, were used in different combinations. The data retrieved was chronologically penned down describing molecular mechanisms of eugenol-induced antiproliferative activity concerning specific cancers. The mechanisms are pictorially presented showcasing the regulation of different epigenetic and transcription factors by eugenol. A scheme of the literature search and selection process is presented in Figure 1.

## 3. Eugenol Characteristics

Eugenol (4-allyl-2-methoxy-phenol; C_10_H_12_O_2_), often known as clove oil, is an aromatic oil derived from cloves (*Syzygium aromaticum*). The eugenol content in clove oil ranges from 70% to 96%. Eugenol is a pale yellowish liquid with an aromatic fragrance that dissolves well in organic solvents and mildly in water. Eugenol has limited chemical stability. Eugenol is prone to oxidation and a variety of metabolic reactions. When taken orally, it is promptly absorbed through several organs and metabolized in the liver. So, eugenol encapsulation seems to be the ideal method for minimizing early absorption, increasing water solubility, and thus boosting activity. Eugenol possesses antibacterial, antifungal, antioxidant, and anticancer properties. Low-dose eugenol appears to have little adverse effects other than local irritation, uncommon allergic responses, and contact dermatitis; however, high-dose eugenol can cause tissue injury and a syndrome of sudden onset seizures, coma, and liver and kidney damage [21].

## 4. Anticancer Potential of Eugenol

An assessment of existing literature clarified that eugenol exerts anticancer activity *via* different yet integrated pathways thus ameliorating the hallmarks of unprecedented cellular proliferation. The suggested techniques can be enlisted as induction of apoptosis, cell cycle arrest, reducing angiogenesis, interplaying dual roles as an oxidant and pro-oxidant, inhibiting inflammation, and stopping cellular invasion and metastasis. Autophagy and necroptosis are also reported in a few studies. The mechanisms involved vary according to the types of cancer dealt with, doses, and time dependence. Studies have also shown the chemopreventive role of eugenol when co-administered with other cytotoxic drugs [22,24,25,26]. Figure 2 illustrates the basic molecular mechanism of eugenol in implementing anticancer effects.

The application of nanotechnology has provided a great platform for improving the therapeutic vigor of many phytoconstituents and eugenol is no exception. Many nanocarriers for eugenol were constructed to increase the therapeutic efficiency such as liposomes, microemulsions, micelles, nanoparticles, magnetosomes, ethosomes, etc. [27]. The improved delivery of nanoengineered phytoconstituents to targeted cancer cells rather than healthy cells has been instrumental in reducing undesirable side effects and resistance to chemotherapeutic agents [28,29]. A detailed discussion of various nanoparticles containing eugenol and their applications in different cancers has been provided in the following sections.

### 4.1. Effect of Eugenol on Breast Cancer

The kind of breast cancer termed triple-negative breast cancer (TNBC) is notoriously deadly. The treatment with cisplatin alone resulted in higher toxicity to normal cells and drug resistance in malignant cells. Combining cisplatin (30 µM) with eugenol (1 µM) potentiated its chemotherapeutic activity by inhibiting aldehyde dehydrogenases (ALDH) enzyme activity, impeding the nuclear factor kappa B (NF-κB) and signaling cascade by reducing binding affinity of the nuclear factor to receptors interleukin 6 (IL-6) and interleukin-8 (IL-8), thus downregulating IL-6 and IL-8 mRNA (messenger ribonucleic acid). In vitro assays on MDA-MB-231, MDA-MB-468, and BT-20 cells and in vivo clarified the apoptotic activity of the eugenol and cisplatin combination via the mitochondrial pathway. Increased Bcl-2/Bax ratio, elevated levels of proapoptotic protein Bax, increased expression of cleaved caspases-3 and -9, cleaved poly (ADP-ribose) polymerase (PARP) on the higher side, and repression of anti-apoptotic protein *B-cell lymphoma 2* (Bcl-2) accounted for the apoptotic potential for the combination of eugenol and cisplatin. The repression of the expression level of matrix metalloproteinase-2 (MMP-2) and matrix metalloproteinase-9 (MMP-9) explained the inhibition of the invasive tendency of the TNBCs by combination therapy [30].

In vivo studies revealed that the combination therapy of eugenol (50.0 mg/kg) and cisplatin (2.0 mg/kg body weight) exhibited potent anticancer activity, utilizing humanized tumor xenograft cells modeled in MDA-MB-231 (*M.D. Anderson-Metastatic Breast 231*) injected into nude mice after the treatment period of 4 weeks. Results revealed the diminution in the development of the tumor is associated with a decrease in the expression of Ki-67 by 95%. Increased apoptosis and angiogenesis of the combination therapy were revealed by the decreased levels of the blood vessel marker cluster of differentiation 31 (CD31). Reduced epithelial-to-mesenchymal transition (EMT) was evident from reduced expressions of N-cadherin and Snail1 and higher E-cadherin expression. Inhibition of pluripotency was evident by reduced expression of biomarker Sox-2 [(sex determining region Y)-box 2] [30].

Eugenol demonstrated anticancer activity by triggering apoptosis in Michigan cancer foundation 7 (MCF-7) (IC_50_: 22.75 𝜇M) and MDA-MB-231 (IC_50_: 15.09 𝜇M) breast cancer cells with increasing ROS levels which inhibited cell cycle at G_2_/M phase, that leads to clastogenesis in vitro. Moreover, it downregulated the proliferation of the cell nuclear antigen (PCNA) associated with deceased mitochondria membrane potential (ΔΨm) and upregulation of Bcl-2 associated X protein (Bax) [31]. 

In a separate study, eugenol induced apoptosis in MCF-7 (EC_50_ value 0.9 mM) adenocarcinoma breast cancerous cells by dose-dependently decreasing the proliferation and cellular viability. It is further associated with a higher level of reactive oxygen species (ROS) and a lower level of ATP and mitochondrial membrane potential (ΔΨm). There was a downregulation in Bcl-2 and Bax expression, but the relative ratio remained unchanged. The release of cytochrome-c and lactate dehydrogenase was also observed at a concentration of eugenol of more than 0.9 mM. Thus, as per observation, eugenol shows non-apoptotic Bcl-2 independent toxicity [32].

Some benzoxazine and aminoethyl derivatives of eugenol were synthesized and their cytotoxicity and cell viability against the MCF-7 cell line were evaluated and reported. Among the various analogs, 6-allyl-3-(furan-2-yl-methyl)-8-methoxy-3,4-dihydro-2H-benzo[e][1,3] oxazine (2) (IC_50_: 21.7 ± 2.90 µg/mL), 6-allyl-3-benzyl-8-methoxy-3,4-dihydro-2H-benzo[e][1,3] oxazine (3) (IC_50_: 26.4 ± 2.68 µg/mL), and 4-allyl-2-(benzylaminomethyl)-6-methoxy phenol (4) (IC_50_: 29.2 ± 2.39 µg/mL) were found to be more potent cytotoxic candidates in comparison to eugenol [33]. 

Eugenol at an IC_50_ value of 1.5 µg/mL is a potent cytotoxic agent which suppressed metastasis and cancer cell migration by downregulating 34.3% matrix metalloproteinase (MMP-9) and 13.7% paxillin mRNA expression levels, respectively, in MCF-7 breast cancer cells [34]. Eugenol-imposed, dose-specific (100–200 µM) preferential cytotoxicity for MCF10A-ras cells excluding ordinary MCF10A cells. Eugenol effectively regulates the mitochondrial pathway of apoptosis as evidenced by the dysregulation of oxidative phosphorylation machinery and reducing the expressions of the transcriptional factors that regulate beta-oxidation of fatty acids. These transcription factors include medium-chain acyl-coenzyme A dehydrogenase, peroxisome proliferator-activated receptor, and carnitine palmitoyl transferase 1 by downregulating c-Myc/peroxisome proliferator-activated receptor-*γ*-coactivator-1*β* (PGC-1*β*)/estrogen-related receptor-*α* (ERR-*α*) pathway [35]. 

Both eugenol and astaxanthin upgraded doxorubicin’s cytotoxic activity with a substantial lowering of its IC_50_ (0.5 μM–0.088 μM) values. Doxorubicin, eugenol (1 mM), and astaxanthin (40 µM) combination synergistically improved the H3 and H4 histone acetylation by increasing histone acetyltransferase (HAT) protein expression, with decreased expressions of forkhead box P3 (FOXP3) protein, and tumor necrosis factor *α* (TNF-*α*) was associated with the elevated mRNA expression level of interferon-*γ* (IFN*-γ*). The combination of doxorubicin and eugenol further decreased the aromatase and epidermal growth factor receptor (EGFR) expression level in contrast to a single treatment of doxorubicin alone. The apoptosis induction by combination treatment of doxorubicin and eugenol was confirmed by increased expressions of caspase 3 and 8 which triggered cell cycle arrest of estrogen receptor-positive MCF-7 cell lines at the S and G_0_/G_1_ phase. These further downregulated the expression level of cytokeratin 7 (CK7). Eugenol induced intrinsic apoptosis through a higher BCl-2/Bax ratio, whereas astaxanthin resulted in the induction of extrinsic apoptosis via both caspase-8 and -3 but downregulated LC3B expression [36]. 

Upon treatment of eugenol in human triple-negative MDA-MB 231 breast cancer cells (4 µM and 8 µM) and HER2 positive SK-BR3 (5 µM and 10 µM) breast cancer cells, cell proliferating and migrating ability was inhibited. MDA-MB 231 cells were used to show the anti-metastatic activity of eugenol, whereas SK-BR3 cells had been utilized to evaluate the antiproliferative and apoptotic effect besides the anti-metastatic activity of eugenol. The biochemical interventions proved increased expressions of cleaved caspases-3, -7, and -9. Inhibition of angiogenesis and metastasis progression was proven by decreased expression of MMP2 and MMP9, respectively. This was collated with unusual increased levels of tissue inhibitor of metalloproteinases-1 (TIMP1) [37].

Eugenol induces apoptosis in Ehrlich ascites carcinoma and MCF-7 breast cancer cell lines momentously under the regulation of the Wnt signaling pathway. Both in vivo and in vitro studies dictated the cytoplasmic degradation of *β*-catenin due to impeded nuclear translocation by phosphorylation of N-terminal residue at ser37. Downregulation of cancer stem cell markers octamer-binding transcription factor 4 (oct4), Notch1 (Neurogenic locus notch homolog protein 1), epithelial cellular adhesion molecule (EpCAM), and CD44 was observed in the stem cells embedded in mammosphere culture. Docking studies in the in-silico binding analysis of eugenol with murine *β*-catenin showed good binding parameters [38]. 

A therapeutic dose (80 μM) of eugenol was shown to cease the proliferating of human epidermal growth factor of receptor 2 (HER-2) positive MCF-10AT cell lines by 32.8%. Insensitivity was observed in the MCF-10A and MCF-7 cell lines which had low levels of HER2 expressions. The major population of MCF-10AT cells showed signs of programmed cell death and remained clustered in the S phase of the cell cycle. Increased dose (180 μM) of eugenol treatment in MCF-10AT cells reduced the protein expressions of Ak strain transforming (AKT), HER2, 3-phosphoinositide-dependent kinase 1 (PDK1), BCL-2, p85, NF-κB, Cyclin D1, and BCL2 associated agonist of cell death (BAD), whereas the Bax, p21, and p27 expressions were augmented. In vivo animal models with breast precancerous lesions treated with 1 mg eugenol inhibited cellular invasion by 30.5% by obstructing HER2/phosphoinositide-3-kinase–protein kinase B (PI3K)-AKT signaling pathways [39].

Molecular hybrids of new sulfonamides were synthesized from eugenol and dihydroeugenol as precursors and cytotoxicity were measured in different cancer cell lines (HT-144, A549, HepG2 cells) including MCF-7 cell. The phenylpropanoid-based sulfonamide (4b) showed antitumor activity against MCF-7 breast cancer cell lines by inducing apoptosis and cell cycle arrest in G_1_/S transition by downregulating cyclins D1 and E [40]. Eugenol (2 µM) induces apoptosis in breast adenocarcinoma cell lines by overexpression of antiapoptotic proteins independent of p53 modulation by downregulating the nuclear kappa factor and E2F family of transcription factors. The anticancer effect was mediated by targeting the E2F transcription factor 1/surviving oncogenic pathway. Upregulation of p21^wafi^ was also evident from the inhibition of cyclin D [41]. In MDA-MB-231 cell lines, eugenol exhibited cytotoxic activity that was dose-dependent, with an IC_50_ value of 2.89 mM during a 48-h incubation period. Studies using *reverse transcription*–*polymerase chain reaction* (RT–PCR) showed that MMP-1, -3, -7, -9, and -11 had lower levels in cells that had been treated with eugenol as compared to cells that had not been treated. In a scratch wound healing experiment, a concentration of 25 µM scratch wound healing agent significantly inhibited cell migration and suppressed the metastasis of MDA MB-231 cells. According to the findings of this research, anti-breast cancer, anti-proliferative, and anti-metastatic effects of eugenol are mediated by targeting matrix metalloproteinase [42]. Eugenol indorses a spurt in reactive oxygen species levels triggering cell-cycle dysregulation, mitochondrial noxiousness, and clastogenesis eliciting apoptosis in breast cancer MCF-7 cell (IC_50_: 22.75 µM) and MDA-MB 231 cells (IC_50_:15.09 µM) in vitro. Oxidative stress in cells abrogates the cell cycle with proliferation cell nuclear antigen (PCNA) down-regulation and decreased mitochondrial transmembrane potential. Cancer cells when exposed to eugenol also showed an increased Bax expression level [31].

Eugenol displayed anticancer action in breast cancer MCF-7 cells at doses ranging from 1 to 4 mM. This activity was shown by a decrease in the intracellular glutathione level, as well as an increase in DNA fragmentation, intracellular H_2_O_2,_ lipid peroxidation, and apoptosis [43].

Separate research found that at concentrations of 5 and 10 𝜇M, eugenol inhibited the growth of HER2-positive (SK-BR-3) and triple-negative (MDA-MB-231) breast cancer cells by lowering Nucleoporin 62, and elevating AKT serine/threonine kinase 1 (AKT), forkhead box O3 (FOXO3a), cyclin-dependent kinase inhibitor (p27), Caspase-3 and -9, cyclin-dependent kinase inhibitor 1A (p21), and apoptosis, and inhibiting PI3K/AKT/FOXO3a pathway [44].

Nanoformulations of eugenol have been reported to possess enhanced chemotherapeutic potential. Chitosan nanoparticles containing eugenol showed a good chemopreventive role against human breast cancer cell lines, MDA-MB 468 (IC_50_: 51 μg/mL) and melanoma cell lines, A-375 (IC_50_: 79 μg/mL) than their non-formulated states [45]. Magnetic field and pH-sensitive targeted drug delivery systems were developed by a group of scientists. They encapsulate eugenol and hesperidin in folic acid-conjugated bovine serum albumin and superparamagnetic calcium ferrite nanohybrids to enhance selective targeting of the hybrid nanoparticles to folate receptor overexpressed breast cancer cells with an encapsulation efficiency of 85.58% and 62.94%, respectively. The in vitro studies on MCF-7 breast cancer cell lines have indicated the reduction in IC_50_ values by 20–30-fold for both of the herbal constituents (eugenol, IC_50_ from 36.27 to 8.75 μg/mL and hesperidin, IC_50_ from 39.72 to 9.08 μg/mL) [29].

### 4.2. Effect of Eugenol on Cervical Cancer

Eugenol is explored for its activities on cervical cancer and is reported to have a promising effect. The combination of methyl eugenol (60 µM) with myricetin (60 µM) synergistically enhanced the cancer cell growth inhibition of cisplatin (1 µM) by inducing strong apoptosis, arresting cells in the G_0_/G_1_ phase of the cell cycle, enhancing ΔΨm, and upregulating caspase-3 activity in the HeLa immortal cervical cell lines [46]. The 3-(4,5-Dimethylthiazol-2-yl)-2,5-diphenyltetrazolium bromide) (MTT) assay, flow cytometry analysis, and dual staining with acridine orange indicated cytotoxicity and apoptosis leading to inhibition of cell proliferation in Henrietta Lacks (HeLa) cervical cell lines with no apoptosis in the controls and normal cells treated with dichloromethane extract of eugenol isolated from *Syzygium aromaticum* [47]. A study showed that eugenol (150 µM) increased the antiproliferative and apoptotic capability of gemcitabine (15–25 mM) in a dose-dependent manner in tested HeLa cervical cell lines in comparison to normal cells. The analysis of genes expressed in the inflammatory process and apoptosis signaling showed downregulation of IL-1*β*, Bcl-2, and cyclooxygenase-2 (COX-2); indicating mediation of an anticancer effect by eugenol *via* apoptosis induction and inhibiting inflammation [24]. Another study showed that a synergistic increase in cancer inhibition was observed for the combination treatment of eugenol (350 µM), cisplatin (0.5–2.5 µM), and X-rays (4–6 Gy) in HeLa cervical cancer cell lines in a time- and concentration-dependent way. Increased expressions of Bax, caspase-3 and -9, cytochrome c, and reduced expressions of COX2 and IL-1*β* unleashed the fact that eugenol causes death *via* apoptosis and mediating anti-inflammatory action [48]. A comparative study of eight phenylpropanoids in combination with 5-fluorouracil (10.5 µM) illustrated eugenol (153 µM) as the most effective entity in minimizing drug-induced toxicity and resistance in normal and cancer cells, respectively. S phase and G_2_-M phase arrest and induction of apoptosis were proposed mechanisms for eugenol’s antiproliferative activity. The combination therapy increased the cell numbers in the G_0_/G_1_ and G_2_/M phase significantly, and the sub-G_1_ phase as well. Involvement of p53, upregulation of caspase-3, dissipation of MMP9, PARP cleavage, etc. are reported in the apoptotic process. In vitro hemolytic activity studies confirm eugenol as less toxic to normal cells [49]. A combination of sulforaphane (6.5–8 µM) and eugenol (200–350 µM) produced an antagonistic and synergistic effect in low and higher doses, respectively. The combination produced synergistic cytotoxicity with gemcitabine (25 mM) at higher sublethal doses as well as in vitro. LD_50_ of EUG and sulforaphane were found to be at 500 μM and 12 μM, respectively. Lowering the expressions of IL-*β*, Bcl-2, and COX-2 were major findings of western blot and RT–PCR studies [50]. Eugenol (50–200 µM) exhibited inhibition of HeLa cell migration. At a maximum concentration of eugenol (200 μM), the migratory rate was reduced by 3.38 ± 1.2 times compared to the control group. Furthermore, it was also discovered that the protein expression of vimentin and Snail-1 was downregulated, whereas that of E-cadherin was upregulated [51].

### 4.3. Effect of Eugenol on Colorectal Cancers

Eugenol induced apoptosis, necrosis, and slowing cell cycle in SW-620 and CACO-2 colon cancer cells after 72 h of treatment, but not in NCM-460 normal cell lines [52]. Eugenol inhibited the mRNA expression of the COX-2 enzyme, a major catalyst of prostaglandin synthesis, specifically PGE2, that plays a key function in producing inflammation and inducing colon carcinogenesis. The LPS-stimulated mouse macrophage HT-29 and RAW264.7 colon cancer cells showed decreased expressions of COX-2 [53]. The human colorectal carcinoma cell lines (HCT-116) showed signs of autophagy and apoptosis after treatment with an active fraction of clove which predominantly contains eugenol. The increased expressions of proteins LC3-II (microtubule-associated protein 1A/1B-light chain 3) and beclin-1 confirmed that eugenol induces autophagy in the colon cancer cells inhibiting their proliferation. Further, the induction of autophagy was synergistically increased following the combination treatment of 3-methyladenine and bafilomycin A1 as autophagy inhibitors with eugenol [54]. Eugenol induced apoptosis in HCT-15 and HT29 human colon cancer cell lines with IC_50_ values of 300 and 500 µM, respectively, via MMP dissipation, activation of caspase-3 and PARP, and upregulation of the p53 tumor suppressor gene. The formation of ROS potentiated apoptotic action by augmenting higher deoxyribonucleic acid (DNA) fragmentation [55].

Eugenol-canola oil or Eugenol-medium chain triglyceride nanoemulsions were taken at a concentration of 750 µM (eugenol) to evaluate the anticancer effect against HTB37 cells; where apoptotic cell death has been observed via ROS generation, cell cycle was arrested at sub G_1_/S phase [56].

Eugenol along with 4-trifluoromethyl benzoic acid (TFBA) was shown to increase cytotoxicity at concentrations of 20.7 and 20.1 µM, respectively, in HCT116 and WiDr cell lines, demonstrating anticancer action [57]. 

### 4.4. Effect of Eugenol on Gastric Cancers

In vivo studies in N-methyl-N’-nitro-N-nitrosoguanidine (MNNG) that induced the rat gastric carcinogenesis model depicted the anti-proliferative and apoptotic activity of eugenol [58,59]. The suggested molecular mechanisms are involved in apoptosis induction, cell cycle arrest, inhibiting angiogenesis, and preventing metastasis progression. The increased expressions of p53, pro-apoptotic proteins, and cell cycle regulatory (inhibitory genes) advocated for the strong apoptotic potential of eugenol. The levels of anti-apoptotic proteins were found to be decreased. The downregulation of the NF-κB family of transcription factors was also profound. The decreased expressions of the matrix metalloproteinase (MMP-2 and MMP-9) with increased levels of tissue inhibitor of metalloproteinases 2 (TIMP-2) gene expression were correlated with the anti-angiogenic potential of eugenol. The anti-apoptotic Bcl-2 protein is upregulated while the pro-apoptotic Bax and caspase-3 proteins have been downregulated after eugenol therapy. The expression of vascular endothelial growth factor (VEGF), vascular endothelial growth factor receptor 1 (VEGFR1), and MMPs are all reduced by eugenol, while that of reversion-inducing-cysteine-rich protein with kazal motifs (RECK) and TIMP-2 is increased, resulting in apoptosis and a decrease in invasion and angiogenesis [59]. EUG inhibits cell proliferation, suppresses NF-κB, promotes cyclin B, cyclin D1, and PCNA expression, and also inhibits expression of the growth arrest and DNA damage-inducible 45 (Gadd45), p53, and p21 proteins in the MNNG-induced male Wistar rat gastric carcinogenesis model [58].

A comparative anticancer study involving capsaicin and eugenol indicated that eugenol can induce apoptosis and inhibit proliferation in human gastric carcinoma cell lines (AGS cells) and was independent of p53 which enhances caspase-8 and caspase-3 expression. In contrast, the apoptotic activity of capsaicin was p53-dependent, and capsaicin-induced the expression of proapoptotic proteins (Bax, caspase-3, and caspase-8) [60].

### 4.5. Effect of Eugenol on Lung Cancer

A study showed that aqueous infusion of clove effectively reduced lung cancer in strain A mice induced by benzo[a]pyrene. The clove infusion administered orally at a dose of 100 mL/mouse/day from the fifth week of benzo[a]pyrene administration and continued up to the 26th week was found to play a potential chemopreventive role due to its apoptogenic and anti-proliferative activities. Further studies showed the upregulation of the expression of pro-apoptotic p53 and Bax proteins and the downregulation of the expression of antiapoptotic Bcl-2 protein in the precancerous stages. Moreover, activation of caspase-3 by clove infusion was manifested from a very early stage of cancer. Clove infusion downregulated the expression of some growth-promoting proteins such as COX-2, Hras, and cMyc [61]. Studies on MRC-5 human embryonic lung fibroblast cells and A549 lung adenocarcinoma cells treated with eugenol (800 µM and 400 µM, respectively) in vitro demonstrated inhibition in cell viability, migration, and invasion. Biochemical findings demonstrated the antiproliferative and antimetastatic effect of eugenol by reducing the PI3/AKT pathway and reduction of MMP-2 [62]. 

The therapy with eugenol considerably reduced the growth of the xenograft tumor and significantly increased the overall survival rate of tumor-bearing mice. On a mechanistic level, eugenol was able to inhibit the expression of p65, which in turn led to a reduction in the expression of tripartite motif-containing protein 59 (TRIM59) protein. The antitumor phenotype induced by eugenol was completely reproduced by the absence of TRIM59 in the cells. It was shown that TRIM59 plays a predominant role in modulating the signaling that occurs downstream of eugenol therapy. Ectopic expression of TRIM59 was eliminated as a result of the tumor suppressive impact of eugenol. Through the suppression of the NF-κB-TRIM59 pathway, eugenol was able to inhibit non-small cell lung cancer [63].

The chemopreventive and antiproliferative role of eugenol utilizing an in vivo mice model has been studied. Downregulation of the WNT/beta-catenin signaling pathway was attributed to the anticancer role of eugenol. The decreased expressions of *β*-catenin-dependent cancer stem cell markers (CD44, EpCAM, Notch 1, and Oct4) advocated for the inhibition of metastatic progression [64].

The 1, 2, 3-triazole-isoxazoline derivatives of eugenol, when tested against A549 cells, showed anti-proliferative activity at 17.32–25.4 µM concentration [65].

### 4.6. Effect of Eugenol on Leukemias

Treatment of eugenol on HL-60 cells induced apoptosis. It was inclusive of fragmentation of DNA and ladders on gel electrophoresis. Generation of ROS led to reduction of mitochondrial membrane potential releasing cytochrome c into the cytosol, reducing levels of antiapoptotic proteins initiating apoptosis, and finally cell death. Eugenol induced apoptosis *via* ROS generation in the different leukemia cell lines like HL-60, U-937, 3LL Lewis, HepG2, and SNU-C5 with IC_50_ values of 23.7, 39.4, 89.6, 118.6, and 129.4 µM, respectively, after 48 h [66]. It was discovered that 3,3′-dimethoxy-5,5′-di-2-propenyl-1,1′-biphenyl-2,2′-diol (bis-eugenol) was most cytotoxic towards HL-60 cells with IC_50_ value of 0.18 mM. At a concentration of 1 mM, eugenol inhibited the mRNA expression of the superoxide dismutases. This inhibition was upregulated by the addition of 5 mM of N-acetyl cysteine or glutathione, indicating the generation of ROS, which induced apoptotic cell death. In RAW 264.7 cells, the LPS (lipopolysaccharide)-stimulated COX-2 gene expression was reduced at 500 µM by bis-eugenol defining its anti-inflammatory action [67].

### 4.7. Effect of Eugenol on Liver Cancer

In a study, eugenol–canola oil nanoemulsions emulsified by starch reduced cell viability in the liver (HB8065) and colon (HTB37) cancer cells with IC_50_ values of 500 µM and 750 µM, respectively, as indicated by MTT assay. RT–PCR and flow cytometry analysis revealed apoptotic cell death via ROS generation. In both cell lines, cells were arrested in sub-G_1_/S phases significantly by eugenol and its nanoemulsion [56].

### 4.8. Effect of Eugenol on Gliomas

Eugenol-loaded chitosan nanopolymers (IC_50_: 7.5 µM) convincingly induce apoptosis and inhibition of metastasis in rat C6 glioma cells. Eugenol nanopolymers inhibit the expression of NF-κB and epithelial to mesenchymal transition (EMT) protein, thereby inducing apoptosis and preventing metastasis, respectively. The drug release efficiency was about 96% at acid pH 2.8 [68]. Eugenol (100–300 µM) stimulated PLC-dependent Ca^2+^ discharge from the endoplasmic reticulum and promoted Ca^2+^ influx, most likely via TRPM8 or protein kinase C-sensitive channels, in DBTRG-05MG human glioblastoma cells. In addition, eugenol mediated apoptosis through a Ca^2+^-independent mitochondrial pathway. The apoptosis induced by eugenol occurs through increasing the production of ROS, reducing ΔΨm, discharging cytochrome c, and activating caspase-9/-3 [69].

### 4.9. Effect of Eugenol on Melanomas

In human melanoma G361 cells, eugenol (0.5–2 mM) was shown to have a dose- and time-dependent connection with inducing apoptosis and activating caspases. Apoptosis induction was validated by Hemacolor stain, MTT assay, Hoechst stain, western blotting, and DNA electrophoresis. Cleavage of PARP and DNA fragmentation factor 45 (DFF45), as well as cleavage of caspase-3 and -6, suggested activation after treatment with eugenol (1 mM). The caspase-6 substrate lamin A was cleaved for complete condensation of DNA during apoptosis [70]. Anti-proliferative activity was shown in primary melanoma cell lines in response to eugenol and its six derivatives. Dehydrodieugenol biphenyls derivatives were reported to reduce by about 40–60% cell growth rate. *O*, *O*′-dimethyldehydrodieugenol inhibited around 70 to 80% of the melanoma cells while the 6,6′-dibromodehydrodieugenol produced almost 100% of tested melanoma cell lines. The growth inhibitory activity of 6,6′-dibromodehydrodieugenol was specific to the melanoma and neuroblastoma cells. The IC_50_ values against GR, Waldenstrom macroglobulinemia (WM), purine nucleoside phosphorylase (PNP), LAN-5, and *Gaslini Institute-LI-neuroblastoma* (GILIN) cell lines were found to be 23, 27, 29, 16, and 19 µM, respectively [71].

Eugenol significantly reduced tumors (nearly 40%) and delayed the time to the endpoint (by 19%) in B16 melanoma xenografts. However, despite there being no signs of invasion or metastasis in the treatment group, 50% of the animals still developed metastases regardless of treatment. Apoptosis induction was seen in the melanoma tumors of the eugenol-treated groups. WM1205Lu cell cycle was arrested at S-phase (40%) along with a concurrent reduction in the G_1_ phase. Downregulation of the E2F family of transcription factors except E2F6 was proved by the complementary DNA (cDNA) array analysis, transient transinfection assays, and gel electrophoretic mobility assays. The B16 melanoma xenograft female B6D2F1 mice model showed a 2.4-day delay in tumor growth and reduces 38% of tumor size on day 15 when compared to the control group, and also prevents tumor metastasis [72].

At a concentration of 0.5 µg/mL, hyaluronic acid-coated dacarbazine–eugenol liposomes have been shown to have 95.08% cytotoxicity, whereas dacarbazine solution exhibited only 10.20% cytotoxicity. Also, the percentage of late apoptotic cells has been discovered to be substantially greater (45.16% vs 8.43%). It was discovered that coated dacarbazine-eugenol liposomes had anti-metastatic activity towards resistant melanoma cell lines, most likely via downregulating survivin [73]. Eugenol induces S cell cycle arrest and apoptosis in the G361 cells in a dose and time-dependent manner (1–2 mM). Nuclear condensation, apoptosis-inducing factor (AIF) as well as cytochrome c release into the cytosol, cleavage of PARP and DFF45, and pro-caspase-3 and -9 downregulation all indicated that eugenol-treated cells had been encouraged to induce apoptosis. Cell cycle arrest was accompanied by a time-dependent reduction in the expression of cyclin-A, -D3, -E, cdc2 cdk2, and cdk4 [74].

### 4.10. Effect of Eugenol on Osteosarcoma Cells

The proliferation of Homo sapiens bone osteosarcoma (HOS) cells was suppressed in a dose- and time-dependent way by eugenol. Over the course of 24 h, the survival rates were 91.7% at 0.5 mM, 83.1% at 1.0 mM, 56.6% at 1.5 mM, 25.3% at 2.0 mM, 13.2% at 5.0 mM, and 8.4% at 10.0 mM. Viability decreased to 84.8%, 53%, 25.3%, and 5% of the control after 8, 16, 24, and 48 h of treatment with 2 mM concentration. Hoechst staining revealed that the nuclei of untreated normal cells were uniformly stained, but those of apoptotic cells were stained in an uneven pattern. Apoptosis was further confirmed by DNA gel electrophoresis in cells treated with 2 mM eugenol, which revealed the characteristic ladder pattern characteristic of apoptotic cells. Western blotting revealed elevated amounts of p53, caspase-3, and cleaved PARP. Osteosarcoma cells treated with eugenol were also shown to cleave lamin A and have a decreased level of DFF-45 in the cytosol [75]. After testing D-glucose-eugenol (0.23 mM) in K7M2 cells, researchers found that it inhibited cell growth and had an anti-proliferative effect in a dose- and time-dependent manner [76]

### 4.11. Effect on Prostate Cancer

Ethanolic extract of *Syzygium aromaticum* (clove) buds containing eugenol showed potent cytotoxicity in the tested adenocarcinoma lines. The IC_50_ value of eugenol for prostate (PC3) adenocarcinoma cell lines after 48 h was found to be 89.44 µg/mL. It was witnessed that radiolabeled (^131^I) eugenol had better uptake by cancer cells [77]. 

The combination of 2-methoxy estradiol and eugenol dictated synergistic anticancer activity against tested prostate cancer cell lines, androgen-responsive LNCaP, and androgen-independent PC-3 and DU 145 with a combination index (CI) score of 0.4. The cell cycle was arrested in the G_2_/M phase (IC_50_ values 0.5 µM for 2-methoxy estradiol and 41 µg/mL for eugenol). Inducing apoptosis through Bcl-2-dependent and independent pathways at dosages lower than the individual agents suggest the relevance of this combination for Bcl-2-resistant prostate cancers [78].

### 4.12. Effect of Eugenol on Skin Tumors

Eugenol demonstrated anticancer activity on 7,12-dimethylbenz[a]anthracene (DMBA)-croton oil-induced skin carcinogenesis on female Swiss albino mice. There was a decrease in the percentage of mice that developed tumors, roughly 42%. Tumors on the skin were much shorter in the eugenol-treated group (0.519 cm) compared to the control group (1.789 cm). Eugenol was orally administered to animal groups before tumor induction. RT–PCR and western block techniques confirmed apoptosis induction, confirming that eugenol treatment resulted in downregulating H-ras, c-Myc, and Bcl-2 expressions along with upregulating Bax, p53, and activating caspase-3 expression in the skin lesions [79]. Eugenol pre-treatment delayed the tumor onset in the 7,12-dimethylbenz[a]anthracene (DMBA) (160 nmol) initiated and 12-*O*-tetradecanoylphorbol-13-acetate (TPA) (8.5 nmol) promoted male Swiss albino mice skin carcinogenesis model. Apoptosis induction was confirmed by TUNNEL analysis. p53 and p21^WAF1^ levels were overexpressed in eugenol-treated mice. The levels of iNOS, COX-2 expression, and pro-inflammatory cytokines (IL-6, TNF-*α*, PGE-2) were reduced. Treatment with eugenol also downregulated the NF-κB signaling cascade [80].

### 4.13. Effect of Eugenol on Oral Cancers

Eugenol present in the hydroalcoholic extracts of *Cinnamomum verum* J. Presl was found effective in preventing the proliferation of SCC-25 oral squamous carcinomas cell lines in vitro. Apoptosis augmentation by decreasing mitochondrial outer membrane potential and arresting cell cycle in S-phase was defined for the antineoplastic potential for eugenol [81]. Eugenol is also reported to inhibit mitochondrial respiration in SCC-4 human oral sarcoma cell lines. This property of eugenol can be focused experimentally to advocate anticancer activity *via* the mitochondrial pathway [82]. Eugenol elicited dose-dependent inhibition of cancer in HSC2 human oral carcinoma cell lines. The major findings were apoptosis induction via caspase and BAK overexpression [83]. Table 1 lists the in vitro studies performed on different cancer cell lines for proving the anticancer vigor of eugenol.

### 4.14. Effect of Eugenol on Ovarian Cancer

Cisplatin (5–10 µM) in conjugation with eugenol (1 µM) was shown to inhibit ovarian cell growth in vitro. Apart from that, it increases Hes1-promoted stemness and apoptosis by repressing Notch-Hes1 signaling, thereby decreasing drug resistance ABC transporter genes in SKOV3 and OV2774 cells [84]. In the same experiment, an in vivo study was incorporated, where the female Nu/J mice xenograft model induced by SKOV3 and OV2774 cells injected as single inoculums were taken. Intramuscular injection with eugenol daily (cat # E51719; Sigma, MO, USA) (50 mg/Kg), cisplatin (cat # 1134357, Sigma, MO, USA) (2 mg/Kg), and a combination of both drugs were administered for 21 days, where ovarian cell growth was found to be restricted drastically, along with increasing apoptosis and Hes1 promoted stemness. The drug resistance ABC transporter genes were found to be decreased too [84].

**Table 1 life-12-01795-t001:** In vitro anticancer studies of eugenol.

Cancer Type	Tested Compound	* Cell Line Used	Effect and Mechanism	IC_50_/EC_50_ Value (µM)/mM	References
Breast cancer	Eugenol (EUG)	MCF7 cells, T47-D cells, MDA-MB-231 cells, and non-tumorigenic MCF 10A cell	↑ Apoptosis, ↓ E2F1/surviving ┴ NF-κB and cyclin D1 oncogenes	2.4 µM	[41]
	EUG	MCF-7 cells	↑ DNA fragmentation, ↓ intracellular glutathione level, ↑ intracellular H_2_O_2_ and lipid peroxidation, ↑ apoptosis	1–4 mM	[43]
	EUG and cisplatin	MDA-MB-231, MDA-MB-468, and BT-20 cells	┴ NF-κB and ALDH	EUG (1.0 μM) and cisplatin (30 μM)	[30]
	EUG	MCF-7 cells	↑ Expression of Bcl-2, ↓ intracellular ATP, ↓ membrane potential, ┴ mitochondrial function, ↓ Cyt-c release and LDH	0.9 mM	[32]
	Benzoxazine and aminomethyl derivatives of EUG (6-allyl-3-(furan-2-yl-methyl)-8-methoxy-3,4-dihydro-2H-benzo[e][1,3]oxazine)	MCF-7 cells	↑ Cytotoxicity	21.7 µM	[33]
	EUG	MDA-MB-231 and MCF-7 cells	↓ Mitochondria membrane potential (ΔΨm), ↓ proliferation Cell Nuclear Antigen (PCNA) level, ↑ Bax overexpression, ↑ DNA damage	15.09 µM in MDA-MB 231 cells 22.75 µM in MCF7 cells	[31]
	EUG	MCF-7 cells	↓ MMP-9 expression, ↓ paxilin gene expression, Suppress metastasis	1 and 1.5 µg/mL after 24 and 48h, respectively	[34]
	EUG	MCF-7, BT-474, SKBR-3, and H-ras oncogene transfected MCF10A (MCF10A-ras) human breast epithelial cells	Dose-dependent selective cytotoxicity in MCF10A-ras cells but not in MCF10A cells, ┴ OXPHOS and FAO, ↓ c-Myc/PGC-1*β*/ERR*-α* pathways, ┴ ROS generation	160–200 µM	[35]
	EUG	MDA-MB-231 cells	↓ mRNA expressions of MMP-1, -3, -7, -9, -11, ┴ cancer metastasis, ↑ antiproliferative action	2.89 mM	[42]
	Doxorubicin (DOX) + EUG/astaxanthin (AST)	MCF-7 cells	EUG and AST augmented DOX cytotoxicity, ↑ caspase 3 ↓ CK7 and LC3BI/II ratio	DOX −0.5 μM DOX + EUG 0.088 μM DOX + AST 0.06 μM	[36]
	EUG	SK-BR-3 and MDA-MB-231	↑ Caspase-3, -7, and -9 expressions, ↓ MMP2 and MMP9 gene expression, ↓ triple-negative and HER2-positive breast cancer metastasis, ↑ anti-metastatic effect	-	[37]
	Hesperidin (HES) and EUG in hybrid nanoformulation	MCF-7 cells, L929 fibroblast cells	FA conjugated carrier targets FA receptor-positive breast cancer cells with higher efficacy; ↑ anti-cancer efficacy of HES and EUG by more than 30-folds.	EUG- 36.27 μg/mL; Hesperidin- 39.72 μg/mL; hybrid nanoformulation- 8.75 μg/mL	[29]
	Syzygium aromaticum essential oil (SAEO) or EUG-loaded chitosan nanoparticles	MDA-MB-468 and A-375 cells	↑ antioxidant activity of SAEO and EUG (IC_50_: 204 and 109 μg mL^−1^, respectively). ↑ anticancer potential when formulated chitosan nanoparticles	79 μg mL^−1^ in (A-375 cells; 51 μg mL^−1^ in MDA-MB-468 cells	[45]
	Molecular hybrids of new sulfonamides with EUG or dihydroEUG (4b)	HepG2, A549, HT-144 MCF-7 cell	↓ Cyclin D1 and cyclin E expression, ┴ cell cycle at G_1_/S transition, ↑ apoptosis in MCF-7 cells	-	[40]
	EUG	HER2 positive (SK-BR-3) and triple-negative (MDA-MB-231) cells	↑ Autophagy by microtubule-associated protein 1 light chain 3, ↑ AKT serine/threonine kinase 1 (AKT), ↓ Nucleoporin 62, ↑ forkhead box O3 (FOXO3a), ↑ cyclin-dependent kinase inhibitor (p27), and Caspase-3 and -9, ↑ cyclin-dependent kinase inhibitor 1A (p21), ↑ apoptosis, ┴ PI3K/AKT/FOXO3a pathway	5, 10 µM	[44]
	1, 2, 3-triazole-isoxazoline derivatives of EUG	MCF-7, MDA-MB-231	┴ Proliferation, ↑ cytotoxicity	17.32–25.94 µM	[65]
Cervical cancer	DCM-EUG extract of *Syzygium aromaticum*	HeLa cells	↑ Cytotoxicity, ↑ apoptosis	200 mg/mL	[47]
	Myricetin, methyl EUG, and cisplatin	HeLa cells	↑ Anti-cancer activity, ↑ apoptosis, ┴ cell cycle, ↓ mitochondrial membrane potential, ↑ caspase-3, ↑ lactate dehydrogenase release	(60 μM methyl eugenol + 1 μM cisplatin) or (60 μM Myricetin + 1 μM cisplatin)	[46]
	EUG, cisplatin, radiation	HeLa cells	↓ Proliferation rate, ↑ LDH release, ↑ caspase-3 and -9 activity, ↑ expression of Bax, ↓ expression of B-cell lymphoma (Bcl)-2, ↑ cytochrome c (Cyt-c), ↓ interleukin-1 beta (IL-1*β*), ↓ cyclooxygenase-2 (Cox-2)	350 µM (EUG), 0.5–2.5 µM (cisplatin) and 4–6 Gy X-rays	[48]
	EUG	SIHA cells	ΔΨm didn’t decrease possibly due to resistance, ↓ PCNA levels, ↑ caspase-3 activation, ↑ Bax overexpression, ↑ DNA damage	18.31 µM	[31]
	Gemcitabine + EUG	HeLa cells	↑ Apoptosis and inflammation, ↓ Bcl-2, COX-2, and IL-1*β*	15–25 mM (Gemcitabine) 150 µM (EUG)	[24]
	Sulforaphane + EUG	HeLa cells	↓ Bcl-2, IL-*β*, and COX-2 expressions, ↑ caspase-3	2.5–8 μM (Sulforaphane) EUG (100–350 μM)	[50]
	5-fluorouracil + EUG	HeLa cells	↑ cytotoxic, ↓ G_2_/M phase, ┴ cell cycle in the S phase and G_1_/G_0_ phase,	316 µM (EUG), 21 µM (5-fluorouracil) or combination (153 µM EUG and 10.5 µM 5FU)	[49]
Colon cancer	EUG	Caco-2, SW-620, and NCM-460 cells	↑ Late-apoptosis and necrosis in Caco-2, ↓ reduce cell proliferation, ↓ G_2_ phase or G_1_ phase of cell cycle	218, 166, and 92 µM, respectively, for the NCM-460, Caco-2, and SW-620 cells, respectively at 24 h	[52]
	EUG	HCT-15 and HT29 cells	↑ Apoptosis, ↓ MMP dissipation, ↑ caspase-3 and polyadenosine diphosphate-ribose polymerase (PARP), ↑ p53 tumor suppressor gene, ↑ ROS generation	300 µM and 500 µM for HCT-15 and HT-29 cell, respectively	[55]
	EUG	Lipopolysaccharide-activated mouse macrophage RAW264.7 cells, HT-29 cells	↓ COX-2 expression in lipopolysaccharide-stimulated macrophage RAW264.7 cells, ┴ mRNA expression of COX-2 in HT-29 cells	0.37 µM	[53]
	EUG (32%) and oleanonic Acid (26%) present in the active fraction of clove	HCT-116 cells	↑ Dose- and time-dependent apoptosis via autophagy mediated by PI3K/Akt/mTOR	113.5 µM	[54]
	4-trifluoromethyl benzoic acid (TFBA) + EUG	HCT116, WiDr cells	↑ Cytotoxicity and anticancer activity	20.7 and 20.1 µM for HCT116 and WiDr cells, respectively.	[57]
	EUG-canola oil or EUG-medium chain triglyceride nanoemulsions	HTB37 cells	Apoptotic cell death via ROS generation, ┴ cell cycle at sub G_1_/S phase	750 µM	[56]
Gastric Cancer	EUG and Capsaicin	AGS cells	↑ Apoptotic activity of capsaicin is p53-dependent, ↑ expression of proapoptotic proteins (Bax, caspase-3 and -8). EUG ┴ cell proliferation and ↑ apoptotic activity which was independent of p53, ↑ caspase-8, and -3 expression	250 μM and 1 mM, respectively, for capsaicin and EUG	[60]
Fibrosarcoma	1, 2, 3-triazole-isoxazoline derivatives of EUG	HT-1080	┴ Proliferation	15.31–18.81 µM	[65]
Gliomas	EUG	DBTRG-05MG human glioblastoma cells	↑ Mitochondrial pathway of apoptosis in a Ca^2+^dissociated manner, ↓ Mitochondrial membrane potential, ↑ ROS production, ↑ caspase-9 and -3, ↑ cytochrome c	100–300 µM	[69]
	EUG-loaded chitosan nanosystem	Rat C6 glioma cells	↓ Expression of NF-κB and epithelial to mesenchymal transition (EMT) protein, ↑ apoptosis	7.5 µM	[68]
Leukemia	EUG	HL60	↑ ROS generated apoptosis, ↑ Mitochondrial permeability transition, ↓ Bcl-2, ↑ cytochrome c release	23.7 µM	[66]
	EUG and 3,3′-dimethoxy-5,5′-di-2-propenyl-1,1′-biphenyl-2,2′-diol (bis-EUG)	HL-60	↑ Cytotoxicity, ↑ apoptosis, ┴ COX-2 gene expression	0.18 mM (bis EUG) and 0.38 mM (EUG)	[67]
Liver cancer	EUG-canola oil or EUG-medium chain triglyceride nanoemulsions	HB8065 cells	↑ Apoptotic cell death via ROS generation, ┴ cell cycle at sub G_1_/S phase	500 µM	[56]
Lung carcinoma	1, 2, 3-triazole-isoxazoline derivatives of EUG	A549 cells	┴ Proliferation	17.32–25.4 µM	[65]
	EUG	Normal mouse fibroblast cells and A549 cells	↑ Cytotoxicity against cancer cells but it is non-toxic against normal cells, ↓ expression of *β*-catenin, ↓ CD44, EpCAM, Oct4, and Notch1 expression, ↑ *β*-catenin and GSK-3*β*, ↑ N-terminal phosphorylated Ser37	5 μM	[64]
	EUG	MRC-5 and A549 cells	↑ Antiproliferative and antimetastatic effects, ↓ phosphate-Akt, ↓ MMP2	800 µM and 400 µM in MRC-5 and A549, respectively	[62]
Melanoma	EUG-related biphenyl (S)-6,6′-dibromo-dehydrodi eugenol	WM266-4, SK-Mel28, LCM-Mel, LCP-Mel, PNP-Mel, A-13443, CN-Mel, GR-Mel cells and SbCl2, NB, GI-LI-N, LAN-5 cells	↑ Cytotoxic, ↑ apoptosis, ↑ caspase activation	16–27 µM	[71]
	Hyaluronic acid-coated dacarbazine-EUG liposomes	SK-MEL-28 and B16F10 cells	↓ E2F1/survivin pathway, ┴ cell cycle at S phase, ↑ apoptosis and cytotoxicity, ↓ migration and proliferation	-	[73]
	EUG	SK-Mel-28 and A2058 cells	↑ caspase-3 activation A2058 cells, ↑ Bax overexpression, ↑ DNA damage	7.201 μM (SK-Mel-28) and 12.17 μM (A2058)	[31]
		G361 cells	↑ Apoptosis, ┴ cell cycle at S phase, ↓ expressions of cyclin A, cyclin D3, cyclin E, cdc2, cdk2, and cdk4, ↑ cleavage of DFF45 and PARP, ↑ caspase-3, and -9	1 mM	[74]
Oral squamous cell carcinoma	EUG	HSC-2 cells	↑ Apoptotic cell death	0.5 mM–2 mM	[83]
	Silver nanoconjugates of EUG	KB cells	↑ Cytotoxicity, ┴ cell cycle at S and G_2_/M phase	2.5–50 µM	[85]
	Hydroalcoholic extract of *Cinnamonium verum* containing EUG	SCC25 cells	↑ Cytotoxicity, ↑ apoptosis, ┴ cell cycle at the S phase	24.71 µM	[81]
Osteosarcoma	EUG	HOS cell	↑ Apoptosis by caspase-3 activation, ↑ expression of the p53 tumor suppressor gene, ↑ cleavage of PARP and lamin ↓DFF-45	1.5 µM	[75]
Ovarian cancer	Cisplatin + EUG	SKOV3 and OV2774 cells	┴ Ovarian cell growth, ↑ apoptosis by ↓ Notch-Hes1 signaling, ↑ Hes1 promoted stemness ↓ drug resistance ABC transporter genes	5–10 µM (Cisplatin) + 1 µM (EUG)	[84]
Prostate cancer	EUG radiolabeled with I^131^	PC3 cells	Cytotoxic, better uptake	89.44 µM	[77]
	EUG + 2-methyl estradiol	LNCaP, PC-3, DU 145 cells	┴ Cell cycle at G_2_/M phase, ↑ Bcl-2-dependent apoptotic cell death, ↑ proapoptotic protein Bax	0.5 µM (EUG) + 41 µM (2-methyl estradiol) in LNCaP cells	[78]

The arrows indicate the increase (↑) or decrease (↓) and inhibition (┴). * NCM-460, normal human mucosal epithelial cells; Caco-2 or HTB37, human colon epithelial adenocarcinoma cells; HCT-15, SW-620, HT-29, and HCT 116, human colon cancer cells; HL-60, human promyelocytic leukemia cells; HB8065, liver cancer cell; MRC-5, human embryonic lung fibroblast cells; AGS, human gastric cancer cells; A549, lung cancer adenocarcinoma cells; G361, SK-MEL-28, and B16F10, melanoma cells; DU 145 and PC-3, androgen-independent prostate cancer cells; LNCaP, androgen-responsive human prostate cancer cells; HOS, human osteosarcoma cells; DBTRG-05MG, human glioblastoma cells; SCC25, HSC-2, oral squamous cell carcinoma cells; KB, oral carcinoma cells.

The antiproliferative activity and the possible involved molecular mechanisms of eugenol-induced apoptosis in various animal models have been enlisted in Table 2. Cancer xenograft models are induced by cell lines and cancer animal model is induced by carcinogens. The anticancer effect of eugenol is based on the downregulation of pro-invasive and angiogenic factors and usually occurs by apoptosis, angiogenesis, and invasion. Recent investigations indicated that eugenol has emerged as a potential contender in both in vitro and in vivo investigations. 

## 5. Discussion

Eugenol at the same time enacts as an antioxidant and pro-oxidant in modulating cellular stress responses [8]. Figure 3 depicts a molecular signaling cascade regulated by eugenol in combating proliferation and apoptosis induction.

It is quite appealing from Figure 3 that eugenol majorly targets the intrinsic mitochondrial pathway triggering apoptosis and associated events in almost all types of cancer cells. Mitochondria is the major tool kit for apoptosis regulation. The degeneracy in mitochondrial membrane potential encouraged by the downregulation of the Bcl-2 family of anti-apoptotic proteins and imbalance in the Bax/Bak ratio persuades the cytosol release of pro-apoptotic proteins like c-Myc, AIF, SMAC/DIABLO and ENDO-G.c-Myc binds to APAF-1, thereby activating the caspases and initiating cell death. The apoptotic inhibitory proteins such as survivin and XIAP are inhibited by SMAC/DIABLO further fueling the caspase activation. AIF promotes DNA condensation and degeneration in a caspase-independent manner ultimately resulting in apoptosis [87]. Cancer is the second major cause of death worldwide, with a fatality rate of six million annually [88]. Adjuvating chemotherapy with chemoprevention by natural agents is gaining thrust due to a reduction in chemotherapy-related side effects [89,90,91]. Eugenol has been declared non-mutagenic and non-carcinogenic by the US Food and Drug Administration (FDA) [18,92]. Eugenol has shown promising results in a variety of cancer studies, including those involving gastric cancer, skin tumors, breast cancer, colon cancer, cervical cancer, and prostate cancer. Reduced plasma membrane size and blebbing, DNA fragmentation, and release of small apoptotic bodies that are phagocytosed by surrounding cells are all signs of programmed cell death known as apoptosis [88,93]. Without the process of apoptosis, the human body would be at a much higher risk for numerous diseases, including cancer, acquired immune deficiency syndrome (AIDS), and other conditions. Apoptosis is an essential function of the human body [88,89]. When combined with certain chemo-inhibitory therapies, eugenol produces a synergistic effect that results in a significant decrease in the toxic effects of such treatments on healthy cells [94]. When it comes to malignancies, breast cancer is the second most prevalent one in women and the fourth leading cause of cancer-related deaths globally [95]. Regulation of mammary epithelial cells in women is achieved by a dynamic balancing act between their proliferation and apoptosis [96]. Eugenol has been instrumental in combating aggressive forms of breast cancers alone or in combination with other antineoplastic entities. Downregulation of the Wnt signaling pathway, cancer stem cell markers oct4, Notch1, EpCAM, and CD44 was a common mechanism encountered in many tested breast cancer cell lines with eugenol. Eugenol also reduced the protein expressions of AKT, HER2, PDK1, BCL-2, p85, NF-κB, Cyclin D1, and BAD whereas the Bax, p21, and p27 expressions were upregulated. In vivo animal models with breast precancerous lesions treated with 1 mg eugenol inhibited cellular invasion by 30.5% by obstructing HER2/PI3K-AKT signaling. Eugenol synergistically checked the viability of cervical cancer cell lines with cisplatin, gemcitabine, and radiations in a concentration and time-dependent way. Increased expressions of Bax, caspase-3 and -9, cytochrome c and decreased expressions of COX2 and IL-1*β* unleashed the fact that eugenol causes death via apoptosis and mediating anti-inflammatory action. S phase and G_2_-M phase arrest and induction of apoptosis had been proposed mechanisms for eugenol’s antiproliferative activity. Involvement of p53, upregulation of caspase-3, dissipation of MMP9, PARP cleavage, etc. are reported in the apoptotic process. MMP dissipation, activation of caspases and PARP, and upregulation of the p53 tumor suppressor gene via the formation of ROS-potentiated apoptotic action by augmenting higher DNA fragmentation were found. Apoptosis induction, cell cycle inhibition, inhibiting angiogenesis, and preventing metastasis progression were reported in studied animal models and gastric cancer cell lines. Angiogenesis inhibition was correlated to increased TIMP-2 gene expressions in experimental cell lines. PI3/AKT is a pathway being targeted along with repression of MMP9 in human embryonic lung fibroblast MRC-5 and lung cancer adenocarcinoma A549 cells. The NF-κB-TRIM59 pathway is also downregulated in studied human non-small cell adenocarcinomas. Eugenol also induced the mitochondrial pathway of apoptosis in a Ca^2+^-dissociated manner in human glioblastoma cell lines. Downregulation of the E2F family of transcription factors except E2F6 was reported in studied malignant melanoma cell lines along with a time-dependent decrease in cyclin A, cyclin D3, cyclin E, cdk2, cdk4, and cdc2 expressions. Radiolabeled eugenol is reported in efficient cancer cell uptake with zero interference of radiolabeled isotope in the anticancer potential of eugenol in tested prostate cancer cell lines. Osteosarcoma cells treated with eugenol were shown to have cleaved lamin A and much less DFF-45 in their cytoplasm. The major findings were apoptosis induction via caspase and BAK over expression in oral carcinoma models in vitro.

## 6. Toxicity of Eugenol

Multiple in vivo studies have shown toxic behavior but no studies have been reported on human subjects [97]. The toxicity of eugenol is dose-dependent [8]. EUG is hazardous because of its prooxidant action [98]. According to some reports, the toxicity of EUG may be traced back to the inactivation of proteins caused by the binding of eugenol to the lysine residues in proteins [99]. Human dental pulp cells are susceptible to chromosomal aberrations (CAs) because eugenol, a known contact allergen in dentistry [97], enters the bloodstream after entering dental pulp tissue [99].

## 7. Challenges and Future Scope

Regardless of the multitude of successes achieved through in vitro and preclinical studies, there is minimal progress in propelling natural phytochemicals to clinical trials. By a thorough examination of in vitro and pre-clinical investigations, the limited efficacy of phytochemical-based therapies is explained: (i) in vitro assays involve direct incubations of cancer cell lines resulting in an acute presentation of phytochemicals that show weighty anti-cancer and anti-proliferative effects at tested concentrations that differ from normal physiological conditions, of oral delivery, even upon consumption of large quantities of the raw or the pure phytochemicals extracts; (ii) the bioavailability problem. Due to how readily the human body processes and eliminates phytochemicals, their presence in the normal human diet creates a bioavailability issue. Phytochemicals have a short biological half-life in vivo, which means they are not persistent in physiological systems and have limited therapeutic effectiveness. Eugenol is poorly water soluble and hence has a reduced bioavailability. It is volatile as well as prone to oxidation. The oral administration of eugenol results in quick plasma and blood concentrations, with a mean half-life of 14.0 and 18.3 h, respectively [98]. After oral administration, it is completely metabolized via sulfate and glucuronic acid conjugation and entirely excreted in urine [100]. Around 0.1 percent of the dosage was found in the urine after being eliminated unmetabolized. The substituted propionic acid production, thiophenol synthesis, double bond migration, and allylic oxidation were among the secondary metabolic processes studied [100]. These pharmacokinetic problems can be resolved by encapsulating eugenol in nanocarriers. The current review of literature also furnishes promising data regarding novel nano-delivery systems of eugenol with underscored enhanced therapeutic prospects [27,101]. Carbapol and Tween 80 nanogel of eugenol showed improved drug delivery analyzed through in vitro studies suggested for future implications in periodontitis [102]. The essential oil fractions of thymus and clove mostly comprise eugenol. Chitosan-coated nanoemulsions of essential oils from thymus and clove showed improved permeation through BBB and were effective in inhibiting multidrug-resistant Gram-negative bacteria and Gram-negative microbes responsible for meningitis and encephalitis [103]. Eugenol potentiated the skin permeation and antioxidant potential of resveratrol in nanoformulations and prevented damage against ionizing radiations [104]. Nanoemulsions of eugenol formulated using Tween 80 and labrasol as surfactant and co-surfactant, respectively, showed good permeation through the skin without nil irritation symptoms as evident from in vivo studies on animal models, serving it as a good option for transdermal drug delivery against skin inflammations [105]. The chitosan-incorporated solid lipid nanoparticles of ofloxacin and eugenol posed good pharmacokinetic profiling along with enhanced bactericidal activity against *Psudomonas aeruguinosa* and *Staphylococcus aureus* in vivo [106]. Nanocapsule suspensions of eugenol proved instrumental in reducing lesions in animal models depicting irritant contact dermatitis in comparison to eugenol application alone [107]. The magnetic nanoparticles of eugenol (IC_50_: 23.34 µg/mL) showed enhanced cytotoxicity against tested human cancer cell lines (glioblastoma, lung, and ovarian carcinoma) in comparison to individual applications of eugenol (100.57 µg/mL). The fabricated magnetic nanoparticles also showed higher inhibition against tested Gram-positive and Gram-negative bacterial strains [108]. 

Much research on nanotechnology interventions in cancer chemotherapy embracing eugenol is in progress. Chitosan-encapsulated eugenol nanoparticles showed better anticancer activity than free eugenol in human melanoma (A-375) and MDA-MB-468 human breast adenocarcinoma cell lines. The IC_50_ value of encapsulated eugenol was reduced to half in comparison to free eugenol-treated cells [45]. Liposomal carriers of eugenol coated with dacarbazine showed enhanced cytotoxicity (95%) against metastatic melanoma cell lines in comparison to dacarbazine alone [73].

Eugenol and piperine loaded on polymeric nanocomposites show enhanced anticancer activity in C666-1 nasopharyngeal cell lines [109]. Folic acid conjugated calcium ferrite nanohybrid formulations of eugenol and hesperidin displayed increased cytotoxic activity in MCF-7 treated breast cancer cell lines by the 30-fold reduction in the IC_50_ values of these natural compounds when treated in free forms. Moreover, the magnetic nanoparticles were found to be pH sensitive, being more drugs released in an acidic environment [29].

## 8. Conclusions

This review aimed to congregate various data available from current scientific literature posing eugenol as a lead natural phytochemical exercising its anticancer potential in different cancers focusing on the intervened molecular mechanisms. It is quite clear that eugenol exhibits remarkable potency to fight against a multitude of cancers. It exerts anticancer activity by induction of apoptosis bracketing its antioxidant and anti-inflammatory role. Evaluation of current literature suggests that eugenol targets multiple signaling pathways for implementing its anticancer effects, the major being MAPK/ERK, JNK/STAT3, WnT/*β*-Catenin pathway, E2F1/survivin, and NF-κB signaling cascades. These pathways are regulated by different proteins, transcription factors, and genes which are major targets of eugenol as indicated by preclinical findings. Upregulation of caspases, proapoptotic proteins, dissipation of mitochondrion membrane potential, increased cleavage of PARP, and misbalancing of the Bax/Bcl-2 ratio strongly suggest apoptotic caliber of eugenol. The overexpression of the p53 tumor suppressor gene in eugenol-treated cancer cell models also advocates the capability of arresting the cell cycle in important transition phases that ultimately lead to cancer cell death. Inhibiting angiogenesis, cell invasion, and migration are essentially portrayed in many in vivo and in vitro models. Eugenol also potentiates cytotoxic activities of frontline anticancer drugs exemplifying 5-fluorouracil, gemcitabine, cisplatin, and doxorubicin. An extensive literature survey also indicates the fruitful applications of nanocarriers for the encapsulation of eugenol for enhancement of its multifaceted therapeutic potential. The current understanding of molecular mechanisms modulated by eugenol by intervening in different signaling pathways could lay down a strong foundation to pierce into the effect of eugenol in altering the bioenergetics of cellular metabolisms associated with cancer pathogenesis, which could rather propel clinical applications of eugenol. Future studies can focus on the potential nanosystems that can eliminate the biopharmaceutical issues posed by showcasing eugenol as a versatile therapeutic agent and reducing its toxicity in vivo.

## Figures and Tables

**Figure 1 life-12-01795-f001:**
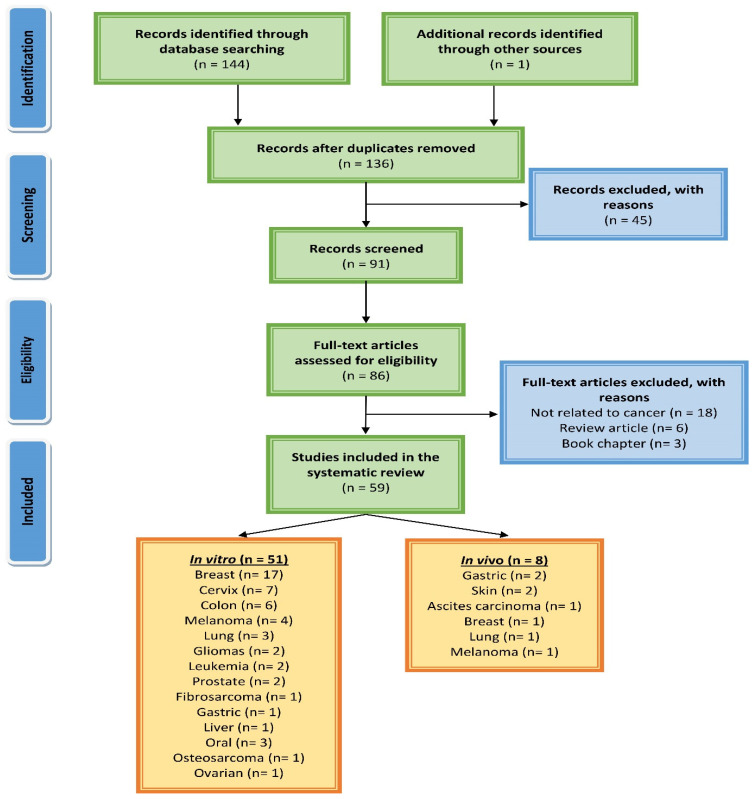
A PRISMA flow diagram showing the process of searching for and selecting studies about eugenol’s molecular mechanisms in cancer.

**Figure 2 life-12-01795-f002:**
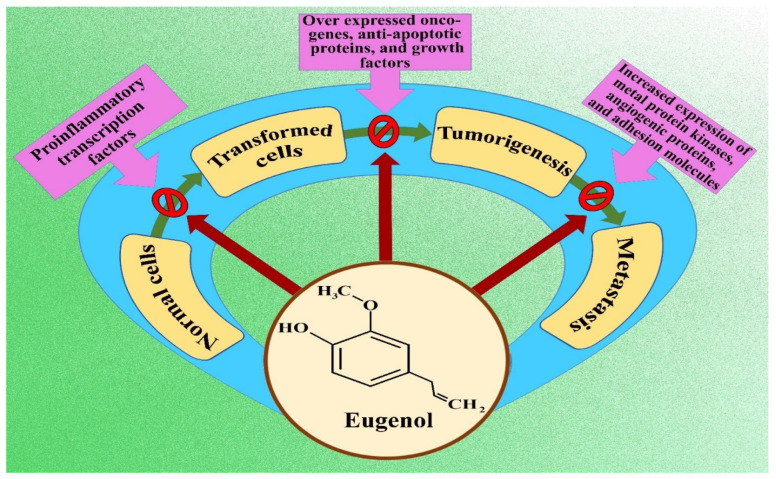
Flow chart indicating molecular mechanism of eugenol inhibiting cancer formation and progression.

**Figure 3 life-12-01795-f003:**
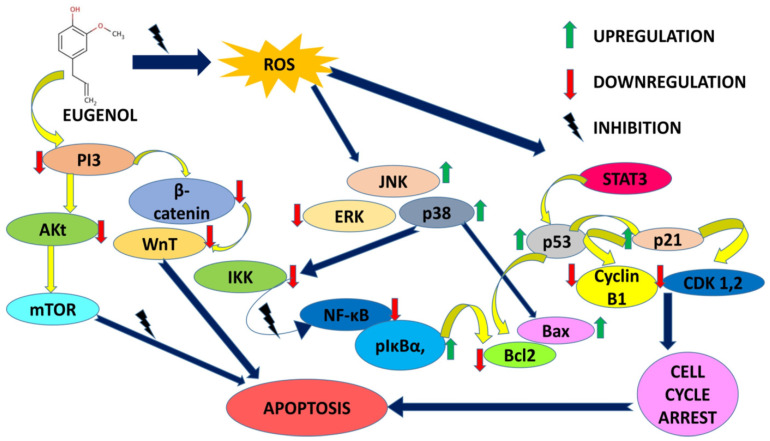
Molecular signaling cascade regulated by eugenol in combating proliferation and apoptosis induction.

**Table 2 life-12-01795-t002:** In vivo studies on anticancer potential of eugenol.

Tested Compound (s) & Cancer Type	Animal Model	Dose of EUG (mg/kg)	Therapeutic Effect	References
Eugenol (EUG) and honey & adenocarcinoma	Ehrlich ascites and solid carcinoma BALB/c mice model	100 mg/kg intraperitoneal (*i.p*.)	┴ Tumor growth (24.35%) in solid carcinoma, ┴ Growth of Ehrlich ascites by 28.88%	[86]
EUG & Breast adenocarcinoma	MDA-MB-231 induced nude mice xenografted human breast tumors	100 mg per kg *i.p*. for 4 weeks.	↓ Tumor growth (66%); ↓ survivin and E2F1 in tumor xenografts	[41]
Clove infusion (EUG) & lung cancer	Benzo[a]pyrene mediated lung carcinoma in strain A mice	aqueous clove infusion of 100 mL per mouse per day orally	↑ Apoptosis, ┴ proliferation, ↑ caspase 3 activation, ↑ Bcl-2/Bax ratio, ↓ COX-2, and some oncogene (cMyc, Hras)	[61]
EUG & gastric cancer	N-methyl-N′-nitro-N-nitrosoguanidine (MNNG) mediated male Wistar rat carcinogenesis model	100 mg/kg body weight three times per week by intragastric route	┴ Cell proliferation, ↓ NF-κB, ↑ cyclin B, cyclin D1, and PCNA expression, ┴ expression of Gadd45, p21, and p53	[58]
EUG & gastric cancer	MNNG mediated gastric carcinomas in male Wistar rats	Starting with MNNG therapy, intragastric eugenol 100 mg/kg body weight, three times per week	↑ Apoptosis, ┴ invasion, ┴ angiogenesis by ↑ expression of TIMP-2 and RECK, ↓ MMPs, VEGF and VEGFR1 expression, ↓ pro-apoptotic Bax and caspase-3 expression, ↑ expression of the anti-apoptotic Bcl-2 protein	[59]
EUG & breast adenocarcinoma	Swiss albino rat with Ehrlich Ascites Carcinoma (EAC) ascetic and tumor xenograft models	25, 50, 100, and 125 mg/kg b.w. for consecutive 14 days	↑ Sub G_1_ from 4.80% to 11.54%; ↓ CSC markers expression	[38]
EUG & Melanoma	B16 melanoma xenograft female B6D2F1 Mice model	125 mg/kg BW twice a week intraperitoneally	↓ Tumor growth by 2.4-day; ↓ tumor size on day 15 (62%); ┴ tumor metastasis	[72]
EUG + cisplatin & triple-negative breast cancer	MDA-MB-231 induced nude mice humanized tumor xenografts and ALDH positive BT-20 induced orthotopic breast tumors	50 mg EUG + 2 mg cisplatin	Cisplatin alone showed a 60% inhibitory effect and combination treatment ↓ tumor growth by 95%	[30]
EUG & ovarian carcinoma	female Nu/J mice xenograft model induced by SKOV3 and OV2774 cells injected as single inoculums	Intramuscular injection daily with eugenol (cat # E51719; Sigma, MO, USA) (50 mg/Kg), cisplatin (cat # 1134357, Sigma, MO, USA) (2 mg/Kg), and a combination of both drugs for 21 days	┴ Ovarian cell growth, ↑ apoptosis, ↑ Hes1 promoted stemness, ↓ drug resistance ABC transporter genes	[84]
EUG & lung cancer	N-nitrosodiethyl- amine induced mouse lung carcinogenesis model	EUG 50 mg/kg body weight of the mouse	Targeted tiny, drug-resistant, and most virulent cancer stem cells, targeting *β*-catenin, ↑ apoptosis ↓ cell proliferation	[64]
EUG & non-small cell lung cancer	NSG immunodeficient mice xenograft model by inoculated subcutaneously TRIM59-deficient H1975 cells into the lower flank	EUG 50 mg/kg b.w. intraperitoneal injection 3 times per week	↓ TRIM59 and p65 expression after treatment, ↑ antitumor effect	[63]
EUG & skin cancer	7,12-dimethylbenz[a] anthracene (DMBA) induced and 12-*O*-tetra decanoylphorbol-13-acetate (TPA) promoted skin cancer in Swiss albino mice	30 mL twice a week for 28 weeks	↑ Apoptosis, ↑ p53 and p21WAF1, ↓ iNOS, COX-2, ↓ TNF-*α*, IL-6, PGE-2	[80]
EUG & skin cancer	DMBA croton oil-induced skin tumor in Swiss mice	1.25 mg/kg body weight orally twice a week	↓ H-ras, c-Myc, and Bcl-2 expression, ↑ Bax, P53 ↑ Caspase-3 expression	[79]

The arrows indicate the increase (↑) or decrease (↓) and inhibition (┴).

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
