# Peer review of "Molecular Mechanisms of Action of Eugenol in Cancer: Recent Trends and Advancement"

_life, 2022, doi:10.3390/life12111795_

Round 1

Reviewer 1 Report

Broad comments:

The review article (life-1969507) entitled “Molecular mechanisms of action of Eugenol in cancer: Recent trends and advancementaims to focus on cellular and molecular mechanisms of eugenol for cancer prevention and therapy. Despite the fact that there are many similar articles on this subject, the work is generally well written.

Introduction

Lines 75 to 83      “However, no previous reviews on eugenol……”

There are really some recent reviews on anticancer properties of eugenol.

Please add.

Methodology for literature search and selection of anticancer studies

The authors provide the criteria for searching for articles, but there is no information about the time range for the article analysis. The article presents a good literature from current research but the time range is difficult to grasp.

Please add.

Did the authors also or only search for combinations of the key words. And which combinations were then used?

Please add a section on eugenol characteristics.

Please also add a section on the molecular mechanisms of action of eugenol in cancer.

Several abbreviations should be written in full words.

I suggest to remove unnecessary figures.

Author Response

Response to Reviewer 1 Comments

Broad comments:

The review article (life-1969507) entitled “Molecular mechanisms of action of Eugenol in cancer: Recent trends and advancement” aims to focus on cellular and molecular mechanisms of eugenol for cancer prevention and therapy. Despite the fact that there are many similar articles on this subject, the work is generally well written.

 Response: We thank the reviewer for their expertise, time, and effort for reviewing our manuscript. We are deeply encouraged by the generous comments regarding the quality of our work. We sincerely appreciate the specific recommendations which we have found extremely valuable while revising our manuscript.

Comment 1. Introduction

Lines 75 to 83      “However, no previous reviews on eugenol……”. There are really some recent reviews on anticancer properties of eugenol. Please add.

 Response: We have included all the recent reviews (References 20-21) as per the suggestions.

Comment 2. Methodology for literature search and selection of anticancer studies

The authors provide the criteria for searching for articles, but there is no information about the time range for the article analysis. The article presents a good literature from current research but the time range is difficult to grasp. Please add.

Response: We thank the reviewer for these constructive comments and have modified the text to clarify that there were no time restraints for this search and the last search was performed in September 2022 (Page 3 lines 98-100).

Comment 3: Did the authors also or only search for combinations of the key words. And which combinations were then used?

 Response: We are indebted to the reviewer for bringing the matter to our attention. Various keywords in different combinations were used (Page 3 lines100-103).

Comment 4: Please add a section on eugenol characteristics.

 Response: We thank the reviewer for this helpful comment and have added a section on eugenol characteristics (page4 lines 112-125).

Comment 5: Please also add a section on the molecular mechanisms of action of eugenol in cancer.

 Response: As per the suggestion we have included a section on the discussion part (page 20 lines 634-644).

Comment 6: Several abbreviations should be written in full words.

Response: We are grateful that the reviewer pointed out this issue. We have included all the full form of the abbreviations throughout the manuscript that we have missed initially.

Comment 7: I suggest to remove unnecessary figures.

 Response: We sincerely appreciate the valuable suggestions which we have found extremely useful while revising our manuscript. We have deleted figure 4 from the manuscript.

Reviewer 2 Report

The authors in this review article provide detailed insight into recent and advanced studies of using eugenol oil in the treatment of different types of cancer. Also, the manuscript reported the Nano formulations of eugenol oil and the different mechanisms of action. The manuscript was well written and the authors gave a lot of important examples of the use of eugenol in the field of oncology.

The manuscript is very interesting however, there are aspects that should be better explored and explained before acceptance for publication.

Comments

1-    Figure 1 is not mentioned in the text

2-    According to figure one I understood that the total reference used in the manuscript was 107 while in the reference part the total number was 108.

3-    According to the author’s guidelines, All Figures, Schemes and Tables should be inserted into the main text close to their first citation and must be numbered following their number of appearances (Figure 1, Scheme I, Figure 2, Scheme II, Table 1, etc.). so please revise all tables and figures.

4-    In figure 2 title, please start with a capital letter.

5-    Figure 4 title, please replace (.) instead of (:).

6-    The abbreviation should be defined the first time they appear and should be added in parentheses after the written-out form.

Author Response

Response to Reviewer 2 Comments

The authors in this review article provide detailed insight into recent and advanced studies of using eugenol oil in the treatment of different types of cancer. Also, the manuscript reported the Nano formulations of eugenol oil and the different mechanisms of action. The manuscript was well written and the authors gave a lot of important examples of the use of eugenol in the field of oncology.

 The manuscript is very interesting however, there are aspects that should be better explored and explained before acceptance for publication.

Response: We thank the reviewer for his/her expertise, time, and effort for reviewing our manuscript. We are encouraged by the generous comments about the quality of our work.

Comments

  • Figure 1 is not mentioned in the text

Response: We are grateful that the reviewer pointed out the fact that figure 1 is not mentioned. We have now mentioned the figure in the text (page 3 lines 107-108).

  • According to figure one I understood that the total reference used in the manuscript was 107 while in the reference part the total number was 108.

Response: We are indebted to the reviewer for bringing the is matter to our attention. There are some necessary corrections in the Figure 1. There is total 110 references but only 59 references were on studies included in systemic review related to eugenol and its cancer application (in vitro and in vivo studies). Other remaining references describes additional information related to general aspects on cancer, eugenol, review articles, book chapters, toxicity, bioavailability, methodology, etc.  

  • According to the author’s guidelines, All Figures, Schemes and Tables should be inserted into the main text close to their first citation and must be numbered following their number of appearances (Figure 1, Scheme I, Figure 2, Scheme II, Table 1, ). so please revise all tables and figures.

Response: As per the suggestion we have made necessary corrections in the manuscript.

  • In figure 2 title, please start with a capital letter.

Response: We have made necessary changes in the Figure 2 title.

  • Figure 4 title, please replace (.) instead of (:).

Response: We have deleted figure 4 as per the suggestions of reviewer 1.

6-    The abbreviation should be defined the first time they appear and should be added in parentheses after the written-out form.

Response: We are grateful that the reviewer pointed out this issue. We have made the necessary changes as suggested.

Round 2

Reviewer 1 Report

All comments were well responded. So, I highly recommend accepting your review.